# An Experimental Investigation on the Cutting Quality of Three Different Rock Specimens Using High Power Multimode Fiber Laser

**DOI:** 10.3390/ma14112972

**Published:** 2021-05-31

**Authors:** Mang-Muan Lian, Youngjin Seo, Dongkyoung Lee

**Affiliations:** 1Department of Future Convergence Engineering, Kongju National University, Cheonan 31080, Korea; lialian2525@gmail.com (M.-M.L.); syjvlfry1004@gmail.com (Y.S.); 2Department of Mechanical and Automotive Engineering, Kongju National University, Cheonan 31080, Korea

**Keywords:** laser cutting, intrusive igneous rocks, multimode fiber laser, kerf width, penetration depth, scanning speed

## Abstract

The laser cutting of rock has been popular recently because of its advantages over traditional rock cutting methods. Several types of research were performed to replace traditional rock cutting techniques with laser cutting. The purpose of this experiment is to observe cutting quality for intrusive igneous rocks using a high-power multimode fiber laser. The cutting quality, in terms of kerf width and penetration depth, resulted from different scanning speeds and was studied and compared. The specimens used in this study were gabbro, granite, and diorite, which are widely applied in the construction industry because of their high compressive strength and beautiful textures. Energy-dispersive X-ray Spectroscopy (EDX) analyses were conducted to observe the chemical content of three different areas, the melting area, the burnt area, and a non-processed area, for each rock specimen. The study of the compositional changes in each area will also go over the cutting quality of each rock specimen at different scanning speeds. According to the experimental results, the kerf widths of the specimens gradually decrease as the scanning speeds increase. The penetration depths into the specimens sharply decrease as scanning speeds increase. From a study of their compositional changes, it is found that the cutting quality for each rock depends on their silica content. This study summarizes that the cutting quality for a rock specimen greatly depends on the scanning speed of the laser cutting.

## 1. Introduction

Due to their useful applications and beautiful textures, igneous rocks are used for many interior decorations and furniture. Moreover, they are used in the heavy construction industry because of their hardness and high strength. Igneous rocks are the most basic type of rocks, formed from the solidification of magma. They are divided into two groups, according to the different magma solidification locations. Intrusive igneous rock is formed from magma solidification beneath the Earth’s upper layer and extrusive igneous rock is formed from magma solidification above the Earth’s surface [1,2,3]. The two types of igneous rocks have different chemical compositions and different structural arrangements; thus, they have different application areas. Extrusive igneous rocks are commonly used as aggregates for concretes. In contrast, intrusive igneous rocks are used for interior decorations, kitchen countertops, and in art galleries [3,4,5]. The most commonly used intrusive igneous rocks are gabbro, granite, and diorite. They are classified according to their different silica contents in the bulk rock composition. Gabbro contains up to 53 wt.% of silica, granite is made up of around 75 wt.% of silica, and 63 wt.% of silica can be observed in diorite, respectively [1,2,3,6,7,8]. Intrusive igneous rocks require neat cutting according to their application areas.

Commonly used conventional rock cutting methods are wire chainsaw cutting, abrasive waterjet cutting, and diamond blade saw cutting [9,10]. However, these conventional rock cutting methods cannot provide the required neat cutting of rocks. The performance of these tools depends on the cutting system as well as the characteristics of the specimens. These types of equipment have many drawbacks, such as producing high levels of noise, unprecise cuts, and large secondary waste. They also require multi-handling and some high maintenance [9,10,11,12].

Compared to the conventional methods, laser-aided manufacturing (LAM) has many advantages, as such it is used in various fields, such as in the automobile industry, construction, aerospace, electronics, and in semiconductors [13,14,15]. Laser cutting is focused on producing the minimum amount secondary waste possible while having a good cutting quality. Laser processing, with the aid of an assistant gas, is well-known for its high cutting speed, high precision, high-quality productivity, low level of noise, low waste production, and ease of automation [13,14,15,16].

In terms of laser processing for rocks, most researchers have studied laser perforation of rocks. Laser perforation provides a high throughput and accurate positioning without leaving any working residue on the material, and it is replacing mechanical perforation in many applications. For most applications, perforation diameters in the range of 40 µm to 800 µm are produced [17,18]. The authors of one paper studied the effects of laser parameters, beam properties, assistant gas efficiency, and rock permeability through the use of a pulsed laser [19]. In another paper, researchers used a high concentration of thermal energy, assisted by laser cutting, to produce a desired small kerf width for rock; they approached this rock cutting technique by relieving the internal strength of the rock [20]. Another researcher described an application in which the combination of laser kerfing and mechanical forces could be used to assist the excavation of hard rock in tunneling and drilling [21]. Next, the authors studied laser destruction and treatment of four different rocks with a CO_2_ laser using up to 5 kW of laser power. The results discussed the effect of power density on different rocks after laser radiation [22]. A further study observed the feasibility of cutting natural granite stones with a CO_2_ laser using a response surface methodology. The main cut quality as well as the economic viability of the process were assessed by estimating the costs associated with the process [23].

However, a systematic approach to the laser cutting quality of intrusive igneous rock has not yet been reported. Moreover, a comparison of three different intrusive igneous rocks after laser cutting has not yet been further studied. In this experiment, the laser cutting quality of three different intrusive igneous rocks will be approached systematically. The effect of laser interaction with the three intrusive igneous rocks, namely, gabbro, diorite, and granite, will be studied. The experiment method will be approached by following the kerf width and penetration depth for each rock. Compositional changes will also be studied in three different regions of each rock. The results obtained from the abovementioned studies will be compared to determine the cutting quality of each rock. This paper is organized as follows. First, the material information, experimental setup, laser parameters, and the experimental procedures are described. Second, the kerf width and the penetration are discussed along with the scanning speeds. Third, EDX analyses of the three different observation areas are performed and discussed. Finally, the concluding remarks of this study are summarized.

## 2. Materials and Methods

Three different rock specimens were prepared with a size of 100 mm × 100 mm × 25 mm, as shown in Figure 1. The three different intrusive igneous rocks had the following thermal and mechanical properties (Table 1) [24].

From the given table, it can be seen that gabbro rock has the highest thermal expansion while granite has the highest compressive strength with the lowest thermal expansion among the three different rock samples.

In this experiment, the cutting was performed using a 9-kW multimode fiber laser, working at a wavelength of 1070 nm. The distance between the cutting nozzle and the specimens was set at 1 mm for all the cutting processes. This study set the laser parameters to be constant except for the scanning speed. The scanning speeds were conducted from 1 to 4.5 mm/s with differences of 0.5 intervals (Table 2).

Cutting tests were performed with the aid of nitrogen assistant gas at a pressure of 13 bars, as shown in Figure 2. The two beds, used as the specimen base holder, were kept 2.2 mm apart from each other. Due to secondary waste products, such as dust and fibers, this experiment used a ventilation duct for safety as well as for sound reducing performance.

## 3. Results and Discussion

### 3.1. Kerf Width

The kerf width of each specimen gradually decreased as the scanning speed increased. As shown in Figure 3, the average kerf width of each specimen at different scanning speeds was lower than 1 mm. Figure 3a shows a comparison of the three specimens distinctively, where granite has the smallest kerf width. Figure 3b shows that the kerf width of gabbro suddenly increased, while the kerf width of diorite abruptly decreased at a 1.5 m/min scanning speed. After a 1.5 m/min scanning speed, the kerf width of the two specimens steadily decreased as the scanning speed increased. On the other hand, the kerf width of granite gently decreased as the scanning speed increased. It could be concluded that kerf widths decrease as scanning speeds increase. Since the laser power used in this study was kept constant for each specimen, a higher laser power usage can only explain the overall results of the small kerf width, which was smaller than 1 mm for each specimen. Although the kerf width of each specimen decreased as the scanning speed increased, very few decrements could occur overall. However, when comparing the three different rocks, a large difference in their kerf widths could be seen, especially in granite, and from both gabbro and diorite. Here, we state that rock composition was the factor that resulted in different kerf widths for each specimen. The compositional effect will be explained in a later part of this paper, according to their EDX analyses.

A one-way analysis of variance (ANOVA) was calculated for the kerf width of each rock specimen at different scanning speeds, as shown in Table 3. The analyses were significant, *F* (2, 21) = 27.41, *p* = 0.000001398. In this case, the kerf width of each rock specimen had a significant difference related to their scanning speeds.

### 3.2. Penetration Depth

It can be seen in Figure 4 that the penetration depth of the specimen decreased as the scanning speed increased. Since gabbro had the lowest percentage of silica among the three specimens, gabbro had the minimum reflectance, so it resulted in the deepest penetration depth. Meanwhile, in granite, which has the highest silica content, the penetration depth was the shallowest due to the high reflection and refraction properties of silica [25,26]. For diorite, even though its kerf width was the largest among the three specimens, the penetration depth results were intermediate. The penetration depth of diorite decreased from 6.5 mm to 2.5 mm for scanning speeds in the range of 1 m/min to 4 m/min. Unlike kerf widths, scanning speed has a great influence on the penetration depth of the laser cutting of rocks. At the same time, the kerf widths of the rock specimens largely depended on the chemical composition of the specimen and the penetration depth of the specimen greatly depended on the chemical composition.

As shown in Table 4, the F statistics are greater than the critical value and the *p*-value is also less than the significance level of 0.05. From this point, the null hypothesis is rejected, and the results showed there was at least one group with a significant difference. This statistical study supports the study results, which note that the penetration depth of each rock specimen has a completely different depth at different scanning speeds.

### 3.3. EDX Analysis

As shown in Figure 5, the EDX analyses were performed in three regions for each specimen. These were the burnt area, non-processed area, and the melting area. The interaction between the high-power laser beam and the specimen resulted in chemical and physical changes to the specimen. The region where the laser bea m directly interacted with the rock resulted in a molten pool and resolidification of the area is called the melting area. The burnt area or the heat-affected zone (HAZ) is the zone of the base material that has not melted but where microstructure and mechanical properties have been affected by the heat generated during laser cutting. The high thermal interaction followed by the rapid solidification of the specimen can cause physical changes in this region, near the cut surface. The non-processed area is the region where there is no interaction between the laser beam and the specimen. These regions were also clearly described in Figure 5a.

As shown in Figure 5b, many different types of chemical elements and fluctuations in their contents can occur in gabbro. Gabbro contains Al, Si, Ca, and Fe compounds in both the non-processed and burnt areas. Chemical decomposition from Ca (OH)_2_ to CaO can occur after laser cutting due to dehydration. This can develop cracks in the burnt area [27]. Here, the melting area is significantly lacking in the Mg, which is the main supporter of both compressive and tensile strengths [28]. During laser cutting, an endothermic reaction occurs in the specimen and the evaporation of the elemental compound takes place. Due to this thermal decomposition reaction, the evaporation of Mg can occur in the melting area, which initially presents in the specimen as a support to specimen strength.

Granite contains the highest amount of silica, with less Fe and Ca, according to Figure 5b. In this specimen, a high content of silica could occur, especially in the burnt area. Since silica has a low thermal conductivity and a high reflectivity, part of the laser beam was reflected and some of it refracted. Meanwhile, all three studied areas lacked Mg. As mentioned previously, this element helped to support the strength of the specimen. Thus, a lack of Mg can reduce the strength of the specimen and it will become easier to perform laser cutting of the specimen. However, the high content of silica in the granite resulted in the shallowest penetration depth and the smallest kerf width, as can be seen in Figure 3 and Figure 4. This result is enough to agree with the statement that laser cutting is also sensitive to the composition of a specimen.

Diorite had nearly similar elemental compounds in its burnt area and melting area. While Fe and Mg could not be found in the non-processed area, the content of Ca was higher than in other areas. In terms of silica content, each region contained nearly the same amount of silica. Since diorite had moderate silica content and Ca content in each area, its kerf width and penetration depth decreased as the laser scanning speed increased.

## 4. Conclusions

This experiment could provide important information about the three different rocks cut using a high-power multimode fiber laser. First, this study notes that the kerf width of a specimen steadily decreases with increased scanning speeds using a high laser power. Second, the penetration depth largely decreases as scanning speed increases. The study confirms that the faster the scanning speed is the shallower the penetration depth is and the smaller the kerf width is. According to the study results, it can be determined that the scanning speed of the laser parameters greatly influences the quality of the cutting of rocks. Since the kerf width of the rocks could be maintained under 1 mm, it is recommended to decrease the scanning speed to achieve deep penetration and a good cutting quality. Finally, it was found that the elemental compounds that made up the specimens also determined the cutting quality of the specimens. The results summarized that it is also necessary to consider the silica content of a specimen when performing laser cutting. To summarize, the highest silica content, in granite rocks, resulted in the smallest kerf width and the shallowest penetration depth compared to gabbro and diorite. This study can classify the cutting quality of gabbro, granite, and diorite as being related to the scanning speed while the other laser parameters were kept constant. As is mentioned earlier, most researchers have studied laser cutting of many different kinds of rocks at the same time. The detailed information of the laser cutting quality of intrusive igneous rocks could not be informed from the previous studies. Since our experiment was mainly focused on intrusive igneous rocks, which are widely used in construction fields, such as home decoration and for kitchen countertops, the detailed information on laser cutting quality of intrusive igneous rocks has been discussed clearly. The comparison of the three rock specimens in terms of the kerf width and penetration depth was more vivid. As can be seen from the statistical analyses, the laser scanning speed had a great influence on kerf width and penetration depth of the specimens. Moreover, the study found that it was essential to consider the chemical composition of the specimens for laser processing. The results from the statistical analyses and the chemical analyses stated that the rock types and the different compositions of each rock specimen could also alter the kerf widths and penetration depths.

## Figures and Tables

**Figure 1 materials-14-02972-f001:**
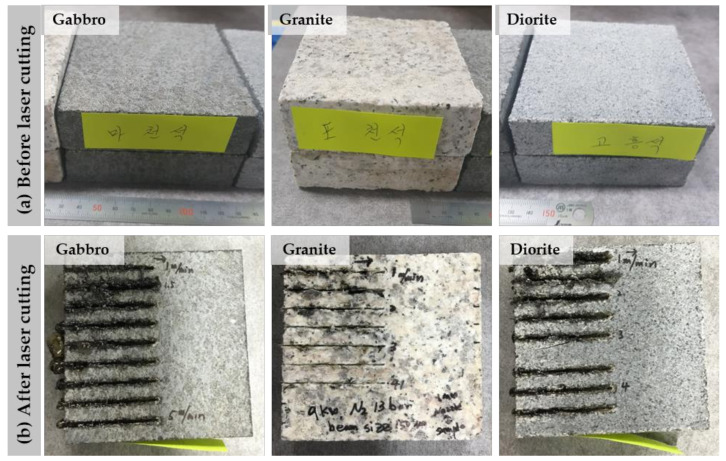
Rock specimens: (**a**) Before laser cutting and (**b**) after laser cutting.

**Figure 2 materials-14-02972-f002:**
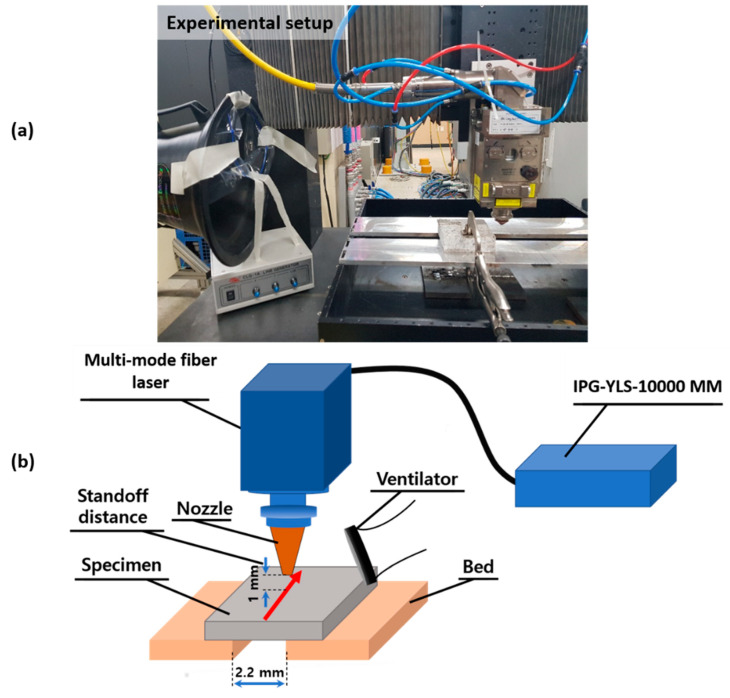
(**a**) Image showing the experimental setup and (**b**) a schematic diagram of the experimental setup.

**Figure 3 materials-14-02972-f003:**
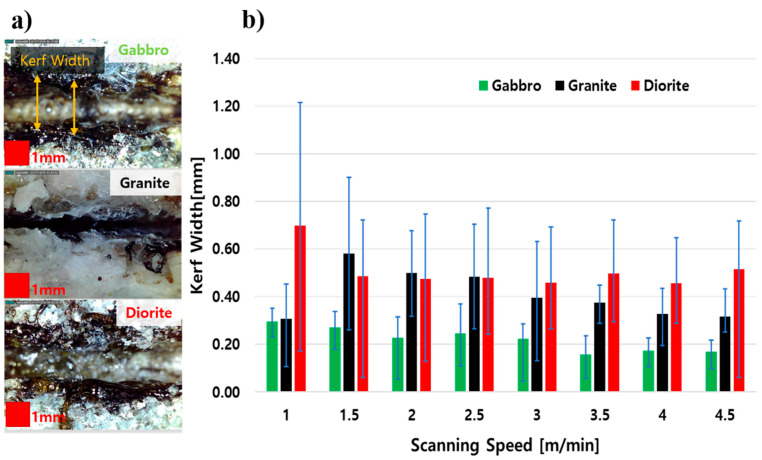
(**a**) The kerf width observation and (**b**) the correlation between kerf width and the scanning speeds of rocks.

**Figure 4 materials-14-02972-f004:**
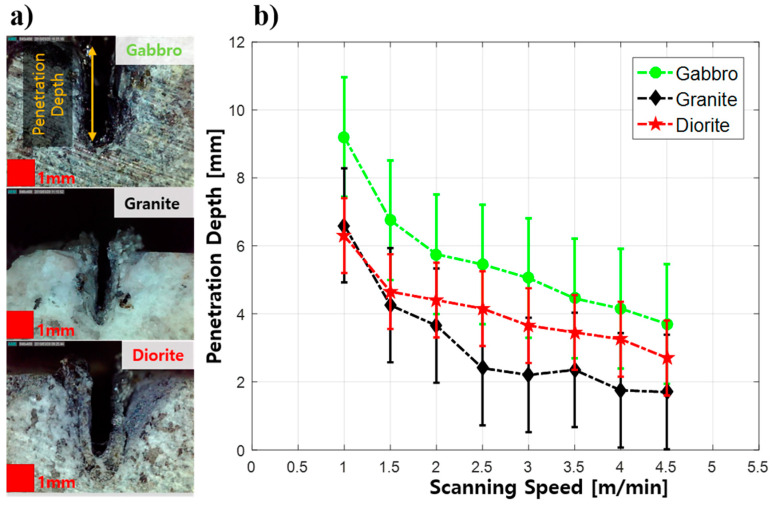
(**a**) The penetration depth observation and (**b**) the correlation between penetration depth and scanning speed.

**Figure 5 materials-14-02972-f005:**
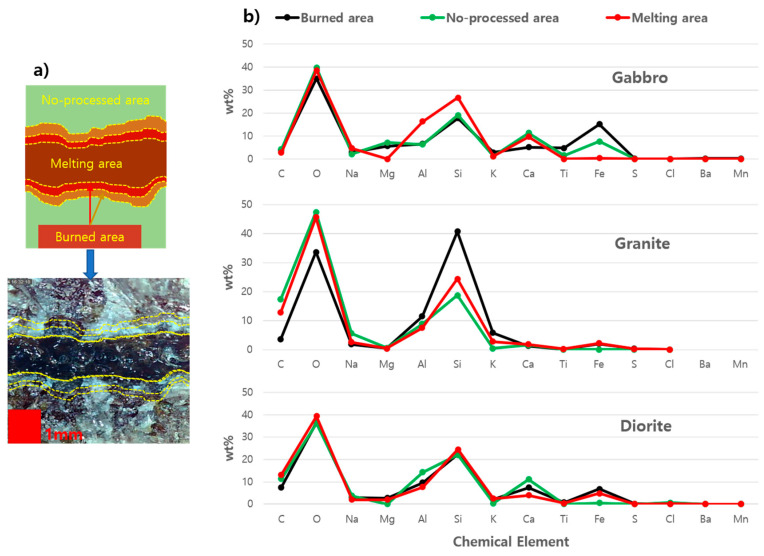
(**a**) Specimen observation tactics diagram and (**b**) EDX analyses of three different regions of gabbro, granite, and diorite.

**Table 1 materials-14-02972-t001:** The typical thermal and mechanical properties of rocks.

Rock Type	Linear Expansion Coefficient (×10^−6^ per Degree Celsius)	Compressive Strength (at Room Temperature and Pressure in MPa)
Gabbro	8 ± 3	223.5
Granite	5.4 ± 1	250
Diorite	7 ± 2	156.5

**Table 2 materials-14-02972-t002:** Laser parameters.

Rock Type	Laser Power [kW]	Wavelength [nm]	Working Distance [mm]	N2 Pressure [bar]	Scanning Speed [mm/s]	Intensity [×10^7^ W/cm^2^]	Line Energy [×10^13^ J/m^3^]
Gabbro	9	1070	1	13	1	5.093	3.056
					1.5	5.093	2.037
					2	5.093	1.528
Granite							
					2.5	5.093	1.222
					3	5.093	1.019
					3.5	5.093	8.731
Diorite							
					4	5.093	7.639
					4.5	5.093	6.791

**Table 3 materials-14-02972-t003:** Statistical analyses for kerf width of each rock specimen related to different scanning speeds.

Descriptive							
Groups	Count	Sum	Average	Variance	Standard Deviation	95% Confidence Interval Mean	
						Lower Bound	Upper Bound
Gabbro	8	3.278	0.4097222	0.00998792	0.035333976	0.374388246	0.445056198
Granite	8	1.758	0.2197222	0.00248668	0.017630529	0.202091694	0.237352751
Diorite	8	4.059	0.5074028	0.00626286	0.027979592	0.479423185	0.53538237
ANOVA							
Source of Variation	Sum of Square	Degree of Freedom	Mean Square	F (statistic)	P-Value	F (critical)	
Between Groups	0.342404248	2	0.171202124	27.41066638	0.000001398	3.466800112	
Within Groups	0.131162247	21	0.006245821				
Total	0.473566718	23					

**Table 4 materials-14-02972-t004:** Statistical analyses for the penetration depths of each rock specimen related to different scanning speeds.

Descriptive							
Groups	Count	Sum	Average	Variance	Standard Deviation	95% Confidence Interval Mean	
						Lower Bound	Upper Bound
Gabbro	8	44.5	5.5625	3.09339286	0.062183125	4.940668747	6.184331253
Granite	8	24.85	3.10625	2.80888393	0.592545771	2.513704229	3.698795771
Diorite	8	32.55	4.06875	1.21924107	0.390391001	3.678358999	4.459141001
ANOVA							
Source of Variation	Sum of Square	Degree of Freedom	Mean Square	F (statistic)	P-Value	F (critical)	
Between Groups	24.50895833	2	12.25447917	5.162303632	0.015014243	3.466800112	
Within Groups	49.850625	21	2.373839286	–	–	–	
Total	74.35958333	23	–	–	–	–	

## Data Availability

Data sharing not available.

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
