# Peer review of "An Experimental Investigation on the Cutting Quality of Three Different Rock Specimens Using High Power Multimode Fiber Laser"

_materials, 2021, doi:10.3390/ma14112972_

Round 1

Reviewer 1 Report

  1. The current study investigates the laser cutting of different rock specimens namely gabbro, granite, and diorite. For this the authors evaluate the kerf width and penetration depth against scanning speeds. The authors found that Kerf width and penetration depths decreased with the increase of scanning speed. Authors also conduced microstructural analysis using SEM to further evaluate the three types of rock.
  2. Please consider reviewing the abstract and highlight the novelty, major findings and conclusions.
  3. The introduction reads well but can be further expanded. The authors should try to add more literature on past studies which did similar work on cutting any types of rock using laser or other conventional cutting methods, mention what they did and what were their main findings then explain how the current study brings new knowledge and difference to the field.
  4. After line 70 please to answer the following question: What is the research gap did you find from the previous researchers in your field? Mention it properly. It will improve the strength of the article.
  5. It is recommended if the authors add some mechanical/thermal properties of the three different types of rocks studies in this literature, this is very important since the laser cutting process can highly influenced by the thermal properties of any material.
  6. Change section 2 title to materials and methods (suggestion)
  7. Section 2 needs further improvement. For instance the authors should clearly add a table summarising the cutting parameters used in this study. Such as the scanning speeds levels and other relevant parameters specifically used in the study.
  8. Line 86 “was set 1 mm in all the cutting processes” why this specific distance was chosen?
  9. I think the authors would have made the study more interesting if they changed more parameters other than the scanning speed.
  10. Please show us pictures of specimens before and after the cutting in section 2.
  11. Please add complete details of all equipment ad test setup used in this study and also add some images. Discuss more about the SEM/EDX analysis as well.
  12. How many cutting tests were conducted for each type of rock and for each laser scanning speed.
  13. Figure 3 is impossible to read this is not acceptable graph type to represent three different sets of data. Please use bar chart and show the range using error bars.
  14. Sections 3.1 and 3.2 there is literally no discussion at all in here. Just describing the graph this is not accepted and must be updated.
  15.  The results are merely described and is limited to comparing the experimental observation. The authors are encouraged to include detailed discussion for each section and critically discuss the observations from this investigation with existing literature

Reviewer 2 Report

A few comments. In general a short and consise paper, but:

- Laser cutting of any material is higly dependent of e.g. the ratio between the beam diameter to nozzle diameter to standoff. So, the paper would benefit from a description of the fiber core diameter, focal distance, intensity profile.

- In fig 2 there is a galvo scanner and a nozzle indicated. Why a scanner when you have a nozzle beneath?

- Why not a fixed optics in this case?

"The study confirms that the faster the scanning speed the shallower is the penetration depth and the smaller is the kerf width."

- This is basically true for all materials, can you explain a little bit more about this for rock material ?

Reviewer 3 Report

Here you can find attached some major revision to improve research quality

Please specify in chapter 2 a table with the process parameters tested (material and laser speed) if replica of experiment have been designed and better describe the method adopted to analyse the results obtained.

Figure 5 and figure 4 are only mean value? What about standard deviation?

In general, the authors should analyse the results with statistical methods as AnoVa in order to assert if there are significant changes between the level of speed tested. Moreover a Tukey analysis could help the authors to identify which level is different from the other ones.

Results section is weak ad obvious, the authors should increase article effort analysing other output parameter as the quality of cutting, the kerf width uniformity or which set produce less burr.

Round 2

Reviewer 1 Report

The authors did not answer point 15, they said they did but checking the revised manuscript does not show any updates or additional discussion. 

the authors need to make sure they answer all questions raised not just saying they did without actually providing an answer.

Please highlight the additions you made for point 15 in different highlight color so I could track the additional discussion you added

Reviewer 2 Report

Dear authors.  Thank you very much for your redesign and clarification of certain points.  Now I understand better that the nozzle is actually mounted onto flying optics.  I'd suggest that you in/outside Fig 2 somewhere add a small text like e.g. "motor driven nozzle", or so.

Otherwise it looks fine.

Reviewer 3 Report

The authors modify the article according to reviewer suggestion however the analisys of results is still weak, in particular referred to the point 3 observation.

The authors answer that beacuse The input parameter in this study was only the scanning speed. they cannot use Anova analysis.

First of all as reports in Table 2 the authors tested both the scanning speed and rock type so an Anova two way could be done. However even if the authors want to analyze only the effect of scanning speed an AnoVa one way is possible and could be very useful the Tukey's range test too.  Looking Figure 4 and 5  it is possible to observe a higher standard deviation and so the risk is that scanning speed no affect the increase or decrease of kerf width or penetration deep, it must be demonstrate by statistical analysis and not by personal point of view. 
